# Combining Coronal and Axial DWI for Accurate Diagnosis of Brainstem Ischemic Strokes: Volume-Based Correlation with Stroke Severity

**DOI:** 10.3390/brainsci15080823

**Published:** 2025-07-31

**Authors:** Omar Alhaj Omar, Mesut Yenigün, Farzat Alchayah, Priyanka Boettger, Francesca Culaj, Toska Maxhuni, Norma J. Diel, Stefan T. Gerner, Maxime Viard, Hagen B. Huttner, Martin Juenemann, Julia Heinrichs, Tobias Braun

**Affiliations:** 1Department of Neurology, Justus-Liebig-University, 35392 Giessen, Germany; francesca.culaj@neuro.med.uni-giessen.de (F.C.); toska.maxhuni@neuro.med.uni-giessen.de (T.M.); norma.diel@ukdd.de (N.J.D.); stefan.gerner@uk-erlangen.de (S.T.G.); hagen.huttner@ukdd.de (H.B.H.); martin.juenemann@neuro.med.uni-giessen.de (M.J.); tobias.braun@lahn-dill-kliniken.de (T.B.); 2NeuroCentrum Wetzlar, Sportparkstrasse 2, 35578 Wetzlar, Germany; mesut.yeniguen@neurocentrum-wetzlar.de; 3Department of Neurology, Lahn-Dill-Kliniken Wetzlar, 35578 Wetzlar, Germany; farzatalchayah@gmail.com; 4Department of Cardiology, Angiology and Critical Care Medicine, Justus-Liebig-University, 35390 Giessen, Germany; priyanka.boettger@googlemail.com; 5Center of Mind, Brain & Behavior (CMBB), 35032 Marburg, Germany; 6Translational Neuroscience Network Giessen (TNNG), 35392 Giessen, Germany; 7Department of Neurology, Kantonsspital Winterthur, 8401 Winterthur, Switzerland; viard.maxime@googlemail.com; 8Department of Radiology, Lahn-Dill-Kliniken Wetzlar, 35578 Wetzlar, Germany; julia.heinrichs@lahn-dill-kliniken.de

**Keywords:** diffusion-weighted image (DWI), brainstem ischemic stroke, coronal DWI, axial DWI

## Abstract

**Background/Objectives:** Brainstem ischemic strokes comprise 10% of ischemic strokes and are challenging to diagnose due to small lesion size and complex presentations. Diffusion-weighted imaging (DWI) is crucial for detecting ischemia, yet it can miss small lesions, especially when only axial slices are employed. This study investigated whether ischemic lesions visible in a single imaging plane correspond to smaller volumes and whether coronal DWI enhances detection compared to axial DWI alone. **Methods:** This retrospective single-center study examined 134 patients with brainstem ischemic strokes between December 2018 and November 2023. All patients underwent axial and coronal DWI. Clinical data, NIH Stroke Scale (NIHSS) scores, and modified Rankin Scale (mRS) scores were recorded. Diffusion-restricted lesion volumes were calculated using multiple models (planimetric, ellipsoid, and spherical), and lesion visibility per imaging plane was analyzed. **Results:** Brainstem ischemic strokes were detected in 85.8% of patients. Coronal DWI alone identified 6% of lesions that were undetectable on axial DWI; meanwhile, axial DWI alone identified 6.7%. Combining both improved overall sensitivity to 86.6%. Ischemic lesions visible in only one plane were significantly smaller across all volume models. Higher NIHSS scores were strongly correlated with larger diffusion-restricted lesion volumes. Coronal DWI correlated better with clinical severity than axial DWI, especially in the midbrain and medulla. **Conclusions:** Coronal DWI significantly improves the detection of small brainstem infarcts and should be incorporated into routine stroke imaging protocols. Infarcts visible in only one plane are typically smaller, yet still clinically relevant. Combined imaging enhances diagnostic accuracy and supports early and precise intervention in posterior circulation strokes.

## 1. Introduction

Brainstem ischemic stroke is one of the most lethal forms of all ischemic strokes and represents approximately 10% of all ischemic brain strokes [1]. The brainstem consists of the midbrain, the pons, and the medulla oblongata. The arterial supply of the brainstem comes mainly through the vertebral arteries and the basilar artery [2]. The white matter of the brainstem binds the cerebral hemispheres with the cerebellum and the spinal cord. Sensory and motor pathways pass through the brainstem, the nuclei of the cranial nerves are located in the brainstem, and multiple important functions are also processed through the brainstem, including respiration, heart rhythm, and alertness. The multitude of systems in a comparatively small area of the brain makes the clinical presentation in the case of ischemic events highly variable. However, diagnosing ischemia in patients with subtle or nonspecific symptoms, such as isolated vertigo, is of the upmost importance for further diagnostics, clinical monitoring of secondary deterioration, and adequate secondary prevention of stroke.

Diffusion-weighted imaging (DWI) is a very reliable technique for the detection of ischemic lesions in the brain [3]. After a few minutes of ischemia, an increased DWI signal in ischemic brain tissue is usually observed. However, it may fail to demonstrate ischemic lesions in patients with acute infarctions, especially if the lesion is smaller than 5 mm in the cranio-caudal direction [4,5,6].

There is increasing evidence that adding coronal to the axial DWI sequences might lead to a higher rate of detection [7,8,9]. Coronal DWI sequences have been implemented in our diagnostic standard and are assessed by the same radiologist for several years; they are also performed on the same MRI device, so our data might strengthen this evidence.

Infarct volume is a critical determinant of clinical severity and prognosis in patients with brainstem ischemic stroke. Studies have shown that larger infarcts are more likely to be associated with severe neurological deficits and worse functional outcomes, as measured by scales such as the NIH Stroke Scale (NIHSS) and the modified Rankin Scale (mRS) [10,11]. However, small infarcts—particularly those in the brainstem—can be clinically significant despite their size, and they may be easily overlooked on imaging if not carefully assessed [12,13]. Ischemic lesions that are visible in only one imaging plane often suggest a limited lesion volume and may reflect smaller, more localized ischemic events [14]. Identifying such imaging patterns and correlating them with diffusion-restricted lesion volume is essential for early diagnosis, risk stratification, and understanding the topographical characteristics of brainstem strokes.

Therefore, in the present study, we aimed to investigate whether brainstem ischemic lesions confined to a single MRI plane are significantly associated with smaller diffusion-restricted lesion volumes, thereby providing quantitative evidence for a frequently observed radiological phenomenon and emphasizing its diagnostic implications. In addition, we sought to evaluate whether coronal DWI, alone or in combination with axial DWI, improves the detection rate of ischemic lesions in the brainstem compared to axial DWI alone. By addressing both volumetric and diagnostic imaging perspectives, this study is intended to contribute to a more nuanced understanding of ischemic lesion characterization in brainstem strokes.

## 2. Materials and Methods

This study is a retrospective analysis conducted at a single-center hospital with a dedicated stroke unit. We analyzed routine data and MRI images of all patients with the clinical diagnosis of brainstem ischemic stroke who underwent MRI with axial and coronal DWI sequences in the period from December 2018 to November 2023. The diagnosis was made either clinically or on the basis of imaging changes. Importantly, clinical diagnoses were based on the expert judgement and experience of neurologists, rather than formal consensus criteria or standardized clinical scales. This approach reflects real-world clinical practice, where experienced neurologists integrate clinical presentation with imaging findings to arrive at a diagnosis.

We assessed patients’ demographics, time from symptom onset to admission, and time from admission to MRI. The symptoms and preexisting stroke risk factors on admission were taken from the electronic patient records and registered. The NIHSS on admission and mRS on admission and discharge were also taken from the electronic record and registered. MRI images were analyzed by a radiologist and a neurologist, both blinded for clinical data and occurrence of DWI lesions in axial or coronal slices, and localization of the lesion (medulla, pons, and midbrain) was documented. Diffusion-restricted lesion volume was assessed using planimetric measurements in both axial and coronal imaging planes. To further estimate diffusion-restricted lesion size, we applied various Euclidean geometric models. These included (1) a spherical model, calculated as 0.75 × planimetric area in the axial plane × maximal height in the coronal plane; (2) the simplified ABC/2 ellipsoid model, defined as (length × width × 5 mm slice thickness)/2, based on axial dimensions; and (3) a modified ellipsoid model incorporating the maximal coronal height, calculated as (length × width × maximal height in the coronal plane)/2.

Approval for this study was obtained from the local institutional review board of the Department of Medicine at Justus Liebig University in Giessen, Germany (approval number: AZ 220/21). The board waived the need for informed consent. The study was registered at www.clinicaltrials.gov (NCT05295862; registered 25 March 2022).

### 2.1. MRI Studies

MRI was performed on a 1.5 T MR system (Philips Ingenia: www.philips.de/healthcare/product/HC781341/ingenia-15t-mr-system) for all patients. A standard protocol was used for all patients, consisting of axial (field of view 180 × 231; matrix 64 × 72; slice thickness 5 mm) and coronal DWI (field of view 97 × 109; matrix 64 × 72; slice thickness 3 mm) sequences. The axial sequences covered the whole brain, whereas the coronal sequences covered only the brainstem and the basal medial sections of the hemispheres.

### 2.2. Statistical Analysis

All statistical analyses were performed using Statistical Product and Service Solutions (SPSS) statistics for Windows (Release 29.0; SPSS, Chicago, IL, USA). Descriptive statistics were employed to summarize data on demographic, clinical, and radiologic characteristics. To assess the normality of the distribution of continuous variables, a Shapiro–Wilk test was used. Normally distributed data were presented by means and standard deviations (mean ± SD) and compared using 2-sided *t* tests, and nonnormally distributed data were presented by median (range) and compared using the Mann–Whitney U test and Wilcoxon signed-rank test, respectively. The significance level was set at alpha = 0.05 with statistical trending in cases of <0.1. Categorical variables were analyzed using the chi-squared test and Fisher’s exact test. Spearman’s rank correlation coefficients were calculated to assess associations between variables. To investigate patient clinical characteristics associated with diffusion-restricted lesion volume, a multivariate logistic regression analysis was performed using diffusion-restricted lesion volumes as a dichotomized at the median as the dependent variable.

### 2.3. Data Availability

The authors declare that all relevant data are included in this manuscript. Individual data can be provided upon reasonable request via the corresponding author.

## 3. Results

### 3.1. Patient Selection

Initially, 1742 patients diagnosed with ischemic stroke between December 2018 and November 2023 were identified. From this cohort, 1536 patients without a diagnosis of brainstem stroke were excluded. Additionally, 72 patients were excluded due to either the absence of MRI scans in two planes or incomplete data. Consequently, 134 patients with brainstem infarction were included in the present study (refer to Figure 1).

### 3.2. Demographics and Clinical Characteristics

In the study sample, 31.3% of patients were female. The median age of the participants was 72 (IQR [interquartile range] 63–81) years, the median time from symptom onset to admission was 0 days (IQR 0–1 days), and the median time from admission to MRI scan was 3 days (IQR of 2–4 days). Diffusion lesions in the brainstem were present in 115 patients (85.8%). In those 115 patients, we found 72 (53.7%) lesions in the pons, 26 (19.4%) in the midbrain, and 21 (15.7%) in the medulla oblongata. In 4 (3.4%) of the 115 patients, we detected lesions in two sections of the brainstem, all located in the midbrain and pons. The demographics and clinical data are depicted in Table 1.

### 3.3. Symptoms and Risk Factors

The median number of clinical signs per patient was four (IQR 3–5), and the median number of risk factors was two (IQR 1–3). Regarding the symptoms reported upon admission, 50% of patients experienced dizziness. Paresis was documented by 48.5% of patients, dysarthria by 37.3%, ataxia by 30.6%, sensory disturbance by 27.6%, eye movement disorder by 20.1%, double vision by 23.9%, and nystagmus by 22.4%. Additionally, 14.2% of patients presented with dizziness as their sole main complaint. Regarding cerebrovascular risk factors for stroke, 73.1% had arterial hypertension, 49.3% had hypercholesterolemia, 29.9% had type 2 diabetes mellitus, and 23.9% had a history of previous stroke. Additionally, 14.9% had atrial fibrillation, and 14.2% were smokers. The symptoms and the risk factors are depicted in Table 2.

### 3.4. Detection of Ischemic Lesions

In nine cases (6.7%), the ischemic lesion was solely detected on axial DWI with one located in the midbrain and eight in the pons. Additionally, in eight cases (6%), comprising four in the midbrain, two in the pons, and two in the medulla oblongata, the ischemic lesion was exclusively identified on coronal DWI (refer to Figure 2 and Table 3).

### 3.5. Planimetric Measurement of Diffusion Lesion Volume in Axial vs. Coronal DWI

The volumetric analysis of diffusion-restricted lesions in the brainstem revealed differences depending on the imaging plane and calculation method. The planimetric measurements varied significantly between the two orientations, reflecting the geometric complexity and potential partial volume effects inherent to brainstem anatomy. It is important to note that these measurements represent the diffusion-restricted lesion volume as seen on DWI, not the final infarct volume. The DWI signal indicates areas of acute ischemic injury but may also include penumbral tissue that could potentially recover, depending on timely reperfusion. There is currently no universally accepted gold standard for estimating infarct size at this stage, and the use of DWI provides a practical but imperfect approximation of the ischemic lesion.

The planimetric volume measured on axial DWI sequences showed a median volume of 366.6 mm^3^ (81.3–1279.6 mm^3^), whereas the planimetric volume on coronal DWI was somewhat lower, with a median of 260.3 mm^3^ (93.0–1457.0 mm^3^). When volumes were estimated using ellipsoidal calculations, the values were notably smaller; using slice thickness alone, the ellipsoidal volume had a median of 124.8 mm^3^ (48.2–276.9 mm^3^). When all measurements were incorporated into the ellipsoidal model, the median volume slightly increased to 139.7 mm^3^ (40.0–443.9 mm^3^) (refer to Table 4).

Finally, the spherical volume estimation yielded a median of 202.8 mm^3^ (41.0–656.0 mm^3^). These results suggest that volumetric estimations vary significantly depending on the imaging plane and the volume calculation method used, with axial planimetric measurements tending to report larger diffusion-restricted lesion volumes compared to coronal imaging and ellipsoidal or spherical estimations.

### 3.6. Sensitivity of Axial vs. Coronal DWI

Axial DWI showed a sensitivity of 84.6% for ischemic lesions in the midbrain, 97.2% for ischemic lesions in the pons, and 90.5% for ischemic lesions in the medulla oblongata. Coronal DWI showed a sensitivity of 96.2% for ischemic lesions in the midbrain, 88.9% for ischemic lesions in the pons, and 100% for ischemic lesions in the medulla oblongata. Axial DWI showed an overall sensitivity of 80.6%, whereas coronal DWI showed an overall sensitivity of 79.9%. Combinations of axial DWI and coronal DWI increased the sensitivity to 86.6% (refer to Table 5).

### 3.7. Correlation Between Imaging Modality and Clinical Severity

A statistically significant correlation was observed between the presence of diffusion lesions on coronal DWI and clinical severity, as measured by the number of complaints, NIHSS, and mRS at admission (Spearman’s correlation; see Table 6). A similar, though less pronounced, correlation was found between the mRS at admission and diffusion lesions on axial DWI. Notably, the presence of diffusion lesions on both axial and coronal DWI demonstrated a stronger correlation with both NIHSS and mRS scores as compared to axial DWI. No statistically significant association was identified between the presence of diffusion lesions (in either axial or coronal DWI) and the time interval from symptom onset to MRI.

### 3.8. Association Between Diffusion-Restricted Lesion Volume and Clinical Status

To assess the relationship between diffusion-restricted lesion volume and the clinical status of patients with brainstem infarction, multiple logistic regression models were conducted using various volume metrics (coronary, axial, ellipsoid, spherical). The clinical status was assessed via the NIHSS at admission. Diffusion-restricted lesion volumes were dichotomized at the median.


**
Association Between NIHSS Score and diffusion-restricted lesion Volume
**


NIHSS scores at admission were significantly associated with diffusion-restricted lesion volume across all volumetric models: (i) Coronary volume: the NIHSS was a significant predictor (OR = 1.260; *p* = 0.003). (ii) Axial volume: a similar significant association was observed (OR = 1.275; *p* = 0.006). (iii) Ellipsoid volume: the NIHSS significantly predicted larger volume (OR = 1.310; *p* < 0.001). (iv) Spherical volume: the strongest association was seen here (OR = 1.388; *p* < 0.001).

2.
**
Spatial Extent of diffusion-restricted lesions and Association with its Volume
**


To explore the spatial extent of infarcts, models including the variable “*Diffusion only in 1 plane*”—representing cases in which the diffusion restriction was visible in only one MRI plane—were analyzed.

A significant inverse relationship was found between this variable and diffusion-restricted lesion volume across all volumetric models. Specifically, (i) for spherical volume, ischemic lesions confined to a single MRI plane were significantly associated with smaller volumes (OR = 0.047; *p*= 0.004). (ii) This pattern was consistent in the ellipsoid volume model (*p* =0.004) and (iii) the coronary volume model (OR = 0.111; *p*= 0.041). (iv) In the axial volume model, the association remained statistically significant (OR = 0.059; *p*= 0.009). These findings suggest that the diffusion-restricted lesions visible in only one MRI plane tend to have significantly smaller volumes. This supports the clinical observation that smaller brainstem infarcts are more likely to be spatially limited and may be detectable in just one imaging plane.

## 4. Discussion

In our retrospective study, using both axial and coronal DWI via MRI led to a higher detection rate of brainstem ischemic stroke within our cohort. A total of 6% of cases revealed the ischemic lesions exclusively on coronal DWI, whereas 6.7% exhibited detection solely on axial DWI. Axial DWI demonstrated a sensitivity of 80.6% in detecting brainstem ischemic strokes, and coronal DWI showed a sensitivity of 79.9%. However, combining axial and coronal DWI increased the sensitivity to 86.6%. Although this figure may initially seem low or insignificant, in individual patients, misdiagnosis can lead to severe consequences for individual patients, including recurrent strokes [15,16,17]. Previous studies showed that ischemic lesions in the brainstem in coronal DWI have been more effectively identified as in axial DWI [7,8,9].

While 1.5 T scanners are widely used in clinical practice and provide reliable diffusion-weighted imaging for stroke detection, it is acknowledged that higher field strengths (e.g., 3 T) may offer an improved signal-to-noise ratio and spatial resolution. This could potentially enhance the visualization of small ischemic lesions, particularly in challenging regions such as the brainstem. However, our study demonstrates that even with 1.5 T imaging, the addition of coronal DWI significantly improves lesion detectability.

The inclusion of coronal DWI can assist the examiners in pinpointing areas of interest [18,19]. Specifically, in our cohort, all ischemic lesions in the medulla oblongata and 96.2% of the ischemic lesions in the midbrain could be visualized using coronal DWI in comparison to standard axial DWI, in which only 90.5% of the ischemic lesions of medulla oblongata and 84.6% of the ischemic lesions of the midbrain could be detected.

Our study’s findings reveal several significant correlations between various clinical and imaging parameters in stroke patients. Specifically, the number of complaints, the severity of the stroke (measured by the NIHSS and mRS at admission), and the presence of diffusion lesions in coronal DWI were significantly correlated. This suggests that patients with more severe strokes and a higher number of initial clinical signs are more likely to exhibit diffusion lesions in coronal DWI. Notably, axial DWI lesions correlated only with the mRS at admission and not with the NIHSS at admission or the number of complaints, unlike coronal DWI. This implies that coronal DWI is more sensitive and relevant in detecting early stroke severity compared to axial DWI.

However, we found no statistically significant correlation between the presence of diffusion lesions (whether in axial or coronal DWI) and the time from symptom onset to MRI. This lack of correlation suggests that the presence of diffusion lesions is not strongly dependent on the timing of the MRI scan postsymptom onset, indicating that diffusion lesions can be detected consistently across different time points after stroke onset and underscoring the robustness of DWI as a diagnostic tool across various stages of poststroke onset.

The present study demonstrates a consistent and statistically significant association between diffusion-restricted lesion volume and clinical severity in patients with brainstem ischemic stroke, as assessed by the NIH Stroke Scale (NIHSS) at admission. Across all volumetric models—coronal, axial, ellipsoid, and spherical—higher NIHSS scores were associated with a greater likelihood of belonging to the larger diffusion-restricted lesion volume group. The volumetric analysis of brainstem infarctions demonstrated notable discrepancies based on the imaging plane and the method of volume calculation employed. Specifically, the planimetric volumes varied significantly across imaging modalities, reflecting the influence of differing geometric assumptions inherent to each technique. This variability highlights a critical limitation in current clinical practice, as the absence of a standardized or gold standard method for ischemic lesion volume measurement complicates both the assessment of lesion size and the comparison of findings across studies. Consequently, the observed inconsistencies underscore the need for consensus on optimal imaging and measurement protocols to enhance the reliability and reproducibility of brainstem infarct evaluation. These findings are consistent with prior research highlighting the prognostic significance of infarct size in supratentorial and infratentorial strokes, in which larger volumes have been correlated with poorer clinical outcomes and higher disability levels [12,20,21].

Importantly, this study also explored the spatial characteristics of ischemic lesions by assessing cases in which diffusion restriction was visible in only a single imaging plane. A robust inverse relationship was observed between this feature and diffusion-restricted lesion volume across all models. Diffusion-restricted lesions visible in a single MRI plane were significantly more likely to be of smaller volume, reflecting their limited spatial extent and potentially more subtle clinical presentation. This observation supports prior reports suggesting that small, focal brainstem diffusion-restricted lesions may be easily overlooked when relying solely on axial DWI, particularly in early stages or when the slice thickness is suboptimal [12,14,22]. Our findings reinforce the need for careful multiplanar evaluation, particularly in patients with clinical signs suggestive of posterior circulation stroke.

While our findings support the diagnostic benefits of supplementing axial DWI with coronal DWI, especially for improving detection in the midbrain and medulla, it is important to consider the practical implications of protocol modification. Incorporating coronal DWI could slightly increase the imaging time and require adjustments to the workflow, which could be a concern in high-throughput stroke centers. However, the coronal DWI sequence is relatively short and can be efficiently integrated into standard protocols, particularly in patients with clinical signs suggestive of brainstem involvement. From a cost-effectiveness perspective, the modest additional imaging time may be outweighed by the clinical and economic consequences of missed diagnoses, which can lead to worse functional outcomes, prolonged hospitalization, and increased long-term healthcare costs. Further cost–benefit analyses and prospective workflow evaluations are needed to determine the optimal implementation strategy across different clinical settings.

Taken together, these results highlight the importance of integrating volumetric and spatial imaging markers in the assessment of brainstem ischemic strokes. The NIHSS, despite its limited sensitivity to posterior circulation strokes, showed a strong correlation with diffusion-restricted lesion volume, supporting its utility even in this challenging anatomical region [23].

Moreover, the finding that smaller diffusion-restricted lesions are often confined to a single imaging plane underscores the diagnostic value of coronal and combined axial-coronal DWI in increasing lesion detection sensitivity. This has practical implications for refining acute stroke imaging protocols and avoiding misdiagnosis or underestimation of brainstem ischemic strokes, which may have serious clinical consequences despite their small size [24].

One limitation of our study is its retrospective design. Furthermore, employing a thinner slice thickness may further enhance the detection rate of brainstem ischemic lesions. Sagittal DWI sequences and thin-slice axial imaging were not included in the standardized protocol for this retrospective cohort. However, future prospective studies that incorporate sagittal and high-resolution axial DWI sequences could help to determine the most effective imaging approach for identifying small brainstem infarcts. Another important limitation is the absence of a universally accepted imaging gold standard for infarct volume estimation in the acute phase. The diffusion-restricted lesion volume on DWI is widely used in clinical practice, yet it may not reflect the final infarct size and may vary depending on acquisition parameters and lesion morphology. Additionally, no formal inter-rater reliability assessment was performed for lesion segmentation, which may introduce some variability into the volumetric measurements. While all measurements were conducted by an experienced rater using a consistent approach, future studies should incorporate blinded, independent ratings and assess inter-observer agreement to strengthen reproducibility. Moreover, difficulties related to brain lesion delineations—such as unclear boundaries of infarct areas and inhomogeneous lesion geometry—pose additional challenges to precise volumetric estimation. Despite these limitations, the observed differences in lesion volumes between axial and coronal DWI orientations remain clinically relevant and consistent with the improved detection rates and better correlation with clinical severity, particularly in the brainstem. However, our study possesses several strengths. The same neurologists established the clinical diagnosis and the same radiologists reviewed the imaging data over the entire study period. Additionally, all participants underwent imaging using a uniform MRI device and standardized protocol, ensuring methodological consistency.

## 5. Conclusions

In conclusion, our cohort achieved a higher detection rate of brainstem ischemic strokes through the combination of axial and coronal DWI. We believe that the additional 2.5 min required to acquire coronal DWI—given its contribution to improved lesion detection and potential clinical impact—warrants its integration into routine stroke imaging protocols. Therefore, coronal diffusion-weighted imaging should be incorporated into standard MRI protocols for patients with suspected stroke in the posterior fossa.

## Figures and Tables

**Figure 1 brainsci-15-00823-f001:**
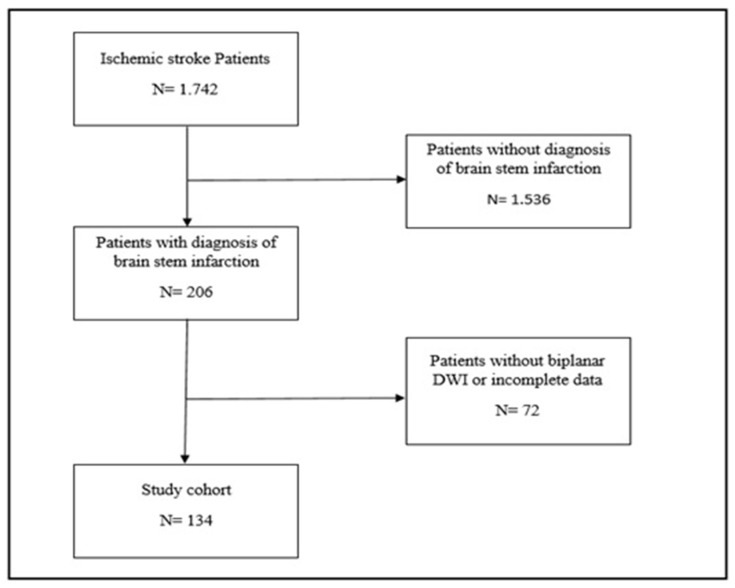
Flow chart of study participants. Initially, 1742 patients diagnosed with ischemic stroke between December 2018 and November 2023 were identified. From this cohort, 1536 patients without a diagnosis of brainstem ischemic strokes were excluded. Additionally, 72 patients were excluded due to either the absence of MRI scans in 2 planes or incomplete data. Consequently, 134 patients with brainstem ischemic strokes were included in the present study.

**Figure 2 brainsci-15-00823-f002:**
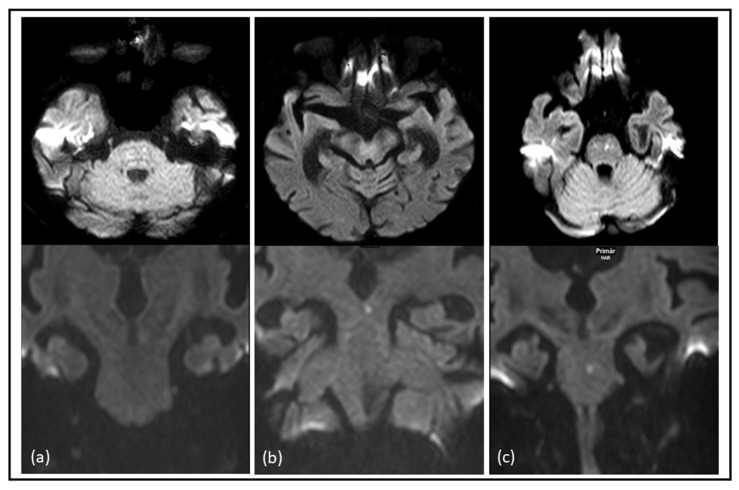
Comparative identification of acute ischemic brainstem stroke on axial and coronal DWI: examples of diffusion-restricted lesions in the brainstem were more easily identifiable on axial diffusion-weighted imaging (DWI) compared to coronal DWI in the pons (**a**), acute diffusion-restricted lesions in the brainstem were more easily identifiable on coronal DWI compared to axial DWI in the mesencephalon (**b**) and diffusion-restricted lesions in the pons were identifiable on both axial and coronal diffusion-weighted imaging (DWI) (**c**).

**Table 1 brainsci-15-00823-t001:** Demographics and baseline clinical Data.

Demographic Characteristics (*n* = 134)	
** *Age (years)* **	72 (63–81)
** *Sex* **	
** * Female *** **	42 (31.3)
** *Clinical characteristics* **	
** *Time of onset until admission (days)* **	0 (0–1)
** *Time from admission until MRI (days)* **	3 (2–4)
** *NIHSS at admission* **	2 (1–4)
** *mRS at admission* **	3 (2–4)
** *mRS at discharge* **	2 (1–3)
** *IVT *** **	19 (14.2)
** *Localization of ischemic lesion in the brainstem* **	
** *Patients without diffusion lesion on MRI *** **	19 (14.2)
** *Patients with diffusion lesion on MRI *** **	115 (85.8)
** * Mesencephalon *** **	26 (19.4)
** * Pons *** **	72 (53.7)
** * Medulla oblongata *** **	21 (15.7)

** *n* (%); median (IQR).

**Table 2 brainsci-15-00823-t002:** Clinical signs and stroke risk factors at admission.

*Clinical Signs at Admission (n = 134)*	
** *Dizziness *** **	67 (50)
** *Paresis *** **	65 (48.5)
** *Dysarthria *** **	50 (37.3)
** *Ataxia *** **	41 (30.6)
** *Sensory dysfunction *** **	37 (27.6)
** *Diplopia *** **	32 (23.9)
** *Nystagmus *** **	30 (22.4)
** *Eye movement dysfunction *** **	27 (20.1)
** *Only dizziness *** **	19 (14.2)
** *Dissociative sensory dysfunction *** **	3 (2.2)
** *Dysphagia *** **	3 (2.2)
** *Crossed motor dysfunction *** **	2 (1.5)
** *Stroke risk factors* **	
** *Arterial hypertension *** **	98 (73.1)
** *Hypercholesterolemia *** **	66 (49.3)
** *Diabetes mellitus type 2 *** **	40 (29.9)
** *Prior stroke *** **	32 (23.9)
** *Prior coronary heart disease *** **	24 (17.9)
** *Atrial fibrillation *** **	20 (14.9)
** *Smoking *** **	19 (14.2)

** *n* (%).

**Table 3 brainsci-15-00823-t003:** Imaging data of DWI-positive brainstem stroke.

	Axial DWI	Coronal DWI	Axial and Coronal Combined
** * Midbrain (n = 26) ** * **	22 (84.6)	25 (96.2)	26 (100)
** * Pons (n = 72) ** * **	70 (97.2)	64 (88.9)	72 (100)
** * Medulla oblongata (n = 21) ** * **	19 (90.5)	21 (100)	21 (100)

** *n* (%).

**Table 4 brainsci-15-00823-t004:** Volumetric analysis.

	Volume in mm^3^
** * Planimetric volume in axial DWI * **	366.6 (81.3–1279.6)
** * Planimetric volume in coronal DWI * **	260.3 (93.0–1457.0)
** * Ellipsoidal volume according to slice thickness * **	124.8 (48.2–276.9)
** * Ellipsoidal volume according to all measurements * **	139.7 (40.0–443.9)
** * Spherical volume * **	202.8 (41.0–656.0)

Median (IQR).

**Table 5 brainsci-15-00823-t005:** Sensitivity of axial and coronal DWI.

	Midbrain	Pons	Medulla Oblongata	Overall Brainstem
** * Sensitivity axial DWI * **	84.6	97.2	90.5	80.6
** * Sensitivity coronal DWI * **	96.2	88.9	100	79.9
** * Sensitivity coronal and axial DWI * **	100	100	100	86.6

**Table 6 brainsci-15-00823-t006:** Spearman’s correlations.

		NIHSS at Admission	mRS at Admission	Number of Complaints
** Coronal DWI Lesion **	ρ (rho)	0.233	0.210	0.184
	Significance (*p*)	0.007 **	0.015 *	0.034 *
** Axial DWI Lesion **	ρ (rho)	0.165	0.177	0.046
	Significance (*p*)	0.057	0.040 *	0.601
**Coronal and Axial DWI Lesion**	ρ (rho)	0.306	0.291	0.091
	Significance (*p*)	<0.001 *	<0.001 *	0.296

ρ (rho): Spearman’s rank correlation coefficient; ** the correlation is 2-sided and significant at the 0.01 level; * the correlation is 2-sided and significant at the 0.05 level.

## Data Availability

The data presented in this study are available on request from the corresponding author. The data are not publicly available due to privacy and ethical restrictions associated with patient information.

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
