# Peer review of "Combining Coronal and Axial DWI for Accurate Diagnosis of Brainstem Ischemic Strokes: Volume-Based Correlation with Stroke Severity"

_brainsci, 2025, doi:10.3390/brainsci15080823_

Round 1
Reviewer 1 Report
Comments and Suggestions for Authors
This is an interesting retrospective analysis in patients with brainstem infarcts showing that combined axial and coronal DWI has superior detection capacity than one-sided imaging. Additional analysis of infarct size and clinical impact of the lesions confirmed what was already known in the literature. Methodology is well designed, conclusions justified, literature well chosen.
A few additional modifications and comments would add value to the paper.
Page3, line 114: 1.5T instrumentation was reported, discuss shortly whether higher Tesla instruments would perform better
Page7, line 192: Estimation of the infarct size, discuss the issue of penumbra, is it included in the estimated lesion area? Diffusion area and infarct lesion are terms for the same descriptive phenomenon?
Page9, Table 6: Interesting table confirming that combined coronal and axial DWI correlate better with clinical features than single reading alone. The statistical method, though, should not be overestimated, because rho-values are rather low and they reflect a weak effect reflecting the effect of a smaller part of the examined cohort rather than a major part, p-values get significant rather because of the high number of associated pairs. This weakness is compensated by the findings described in lines 238-257.
Reviewer 2 Report
Comments and Suggestions for Authors
This manuscript addresses an important and under-recognized challenge in acute stroke diagnostics—under-detection of small brainstem infarcts due to limitations in conventional axial DWI imaging. The authors convincingly demonstrate that supplementing axial DWI with coronal DWI modestly increases sensitivity and correlates better with clinical severity measures, particularly in the midbrain and medulla. The study is methodologically sound, with a well-defined patient cohort, standardized imaging protocol, and appropriate statistical analyses. However, there are a few areas where the manuscript could be strengthened. First, while the volumetric data are comprehensive, the absence of a clear imaging gold standard or inter-rater reliability assessment limits the interpretability of the volume comparisons across planes. Second, although the conclusion advocates for integrating coronal DWI into routine stroke protocols, the added clinical value must be balanced against imaging time, workflow constraints, and cost—issues not adequately addressed in the discussion. Third, the authors could better contextualize their findings by comparing their sensitivity gains to those achieved in other imaging planes like sagittal or using thin-slice DWI. Finally, some data presentations (e.g., Tables 4–6) could benefit from clearer clinical framing to guide interpretation. Despite these limitations, the paper provides solid evidence supporting the diagnostic utility of coronal DWI and is a meaningful contribution to improving stroke imaging protocols. I recommend minor to moderate revision to address the noted issues and enhance clinical relevance.
Reviewer 3 Report
Comments and Suggestions for Authors
The study valuably addresses diagnostic gaps in brainstem stroke imaging. To solidify its impact, methodological refinements, resolution of data inconsistencies, and balanced discussion of clinical integration would be beneficial. The addition of coronal DWI shows promise, though broader implications merit further contextualization.
- The inclusion of patients diagnosed clinically or via imaging might introduce heterogeneity, as clinical diagnosis of brainstem strokes can be challenging. Further details on how clinical diagnoses were standardized (e.g., using consensus criteria) would enhance methodological transparency.
- The overall sensitivity rates in Table 5 (80.6% for axial, 79.9% for coronal) seem inconsistent with the 85.8% detection rate in Section 3.2. Please clarify how "overall sensitivity" was derived would resolve this apparent discrepancy.
- While statistically significant, the modest Spearman’s ρ values (0.18–0.23) for coronal DWI correlations with NIHSS/mRS suggest these relationships are clinically nuanced. A more cautious interpretation of their strength might better reflect the data.
